# Melatonin, temazepam and placebo in hospitalised older patients with sleeping problems (MATCH): a study protocol of randomised controlled trial

Fiona Stenveld,[1] Sjanne Bosman,[1,2] Barbara C van Munster,[1,3]
Sara J Beishuizen,[4] Liesbeth Hempenius,[2] Nathalie van der Velde,[4]
Nynke Smidt,[1,5] Sophia E de Rooij[1,4]

FS and SB contributed equally.

**Correspondence to**
Ms. Sjanne Bosman;
s.bosman@umcg.nl

## ABSTRACT

**Introduction** Hospitalised older patients frequently suffer from inadequate sleep, which can lead to patient distress and delayed recovery from acute illness or surgical procedure. Currently, no evidence-based treatments exist for sleeping problems in hospitalised older patients. Benzodiazepines, such as temazepam, are regularly prescribed by physicians, although they have serious side effects; for older patients in particular. Melatonin is proposed as a safe alternative for sleeping problems in hospitalised older patients, but the efficacy of melatonin is unclear in this population. Therefore, the aim of this study is to investigate the effects of melatonin and temazepam compared with placebo on sleep quality among hospitalised older patients with sleeping problems.

**Methods and analysis** This study is a multicentre, randomised, placebo-controlled trial. A total of 663 patients will be randomised in a 1:1:1 fashion to receive either melatonin (n=221), temazepam (n=221) or placebo (n=221). The study population consists of hospitalised patients aged 60 years and older, with new or aggravated sleeping problems for which an intervention is needed. The primary outcome is sleep quality measured with the Leeds Sleep Evaluation Questionnaire (LSEQ). Secondary outcomes include sleep parameters measured with actigraphy and medication-related adverse effects.

**Ethics and dissemination** This study was approved by the Medical Ethics Committee of the Academic Medical Centre Amsterdam, (No 2015_302). Study findings will be disseminated through presentations at professional and scientific conferences and publications in peer-reviewed journals.

**Trial registration number** NTR6908; Pre-results.

## INTRODUCTION
### Background

Hospitalised older patients frequently experience sleeping problems due to medical, psychological or environmental disturbances. The prevalence of sleep problems among older patients in medical and surgical wards varies between 30% and 65%.[1–3] Apart from the feeling of tiredness, insufficient sleep

### Strengths and limitations of this study

► The results of this study will provide knowledge relevant for clinical practice and fill gaps in current guidelines concerning sleep problems in hospitalised older patients.
► To optimise the generalisability of the findings, this is a multicentre study with participants from different wards and minimal exclusions criteria.
► This study investigates, next to the effects, also the adverse events of melatonin and temazepam compared with placebo.
► Patients with cognitive impairment are excluded due to ethical considerations, even though these vulnerable patients may benefit the most from the study outcomes.

has substantial consequences for hospitalised patients. This includes delayed recovery from acute illness or surgical procedures because of its immunosuppressive effect,[4 5] risk of delirium and falls[6–9] and disturbances in metabolic and endocrine functioning.[10 11]

Thus, adequate treatment of sleeping problems is necessary. Today, it is still unclear how sleeping problems should be treated in the hospital. Non-pharmacological interventions are advised as first choice therapy in hospitalised patients,[12–14] but there is limited evidence to support the effectiveness of non-pharmacological interventions.[15] With regard to pharmacological interventions, there is insufficient evidence to support its effectiveness for improving quality or quantity of sleep in hospitalised patients suffering from poor sleep.[16]

Currently, up to 29% of older hospitalised patients with sleeping problems receive benzodiazepines,[17 18] despite a recent meta-analysis of risks and benefits for short-term treatment of insomnia, which stated that benzodiazepines have little effect and may not outweigh

the risks of adverse events, especially in a high-risk older population.[19] Older patients have a higher risk of adverse drug events from benzodiazepines due to altered pharmacokinetics and pharmacodynamics. Common adverse drug events include in-hospital falls, delirium, aspiration and respiratory depression.[20 21] Furthermore, use of benzodiazepine during hospitalisation can lead to prolonged use at home, which might promote chronic use and substance-dependency.[22]

The neuroendocrine hormone melatonin has been proposed as a safer alternative to improve sleep quality because of its limited side effects.[8 23–25] Melatonin, produced by the pineal gland at night, is well known to influence circadian rhythm and sleep.[26] Two randomised control trials (RCTs) investigated the effectiveness of melatonin compared with placebo in older community dwelling persons with chronic insomnia and found that melatonin improved sleep quality significantly.[27 28] However, studies on the effectiveness of melatonin for sleeping problems in acute hospitalised older patients are scarce. Three RCTs have been performed in adult patients on the intensive care unit,[29–31] showing inconsistent results; one out of three reported a positive effect of melatonin.[30] All three studies were underpowered and none specifically assessed older patients with predefined sleep related problems.

Only one small RCT investigated the effectiveness of melatonin on a general medical ward in adults, which resulted in beneficial effects for melatonin compared with placebo on the sleep onset latency and patient satisfaction.[32]

The overall aim of this study is to establish an evidence-based pharmacological treatment for hospitalised older patients with sleeping problems. In this study, we will first hypothesise that melatonin and temazepam are more effective than placebo on sleep quality among hospitalised older patients with sleeping problems and second melatonin is more effective than temazepam.

## Objectives

The primary objective is to investigate first the effects of melatonin and temazepam compared with placebo among hospitalised older patients on self-reported sleep quality. Second, to investigate if melatonin is more effective than temazepam.

The secondary objectives are to investigate the effects of melatonin, temazepam and placebo on:
1. Other self-reported sleep parameters: getting to sleep, awakening from sleep, behaviour following wakefulness.
2. Objective sleep parameters: reduction in sleep onset latency, sleep efficiency, number and duration of wake bouts, time awake after sleep onset.
3. Adverse events related to study medication: incidence of delirium, cognition, number of falls, complications.
4. Length of hospital stay in days.
5. Time needed to reach improvement in self-reported sleep quality.
6. Proportion of good nights of sleep during study period.

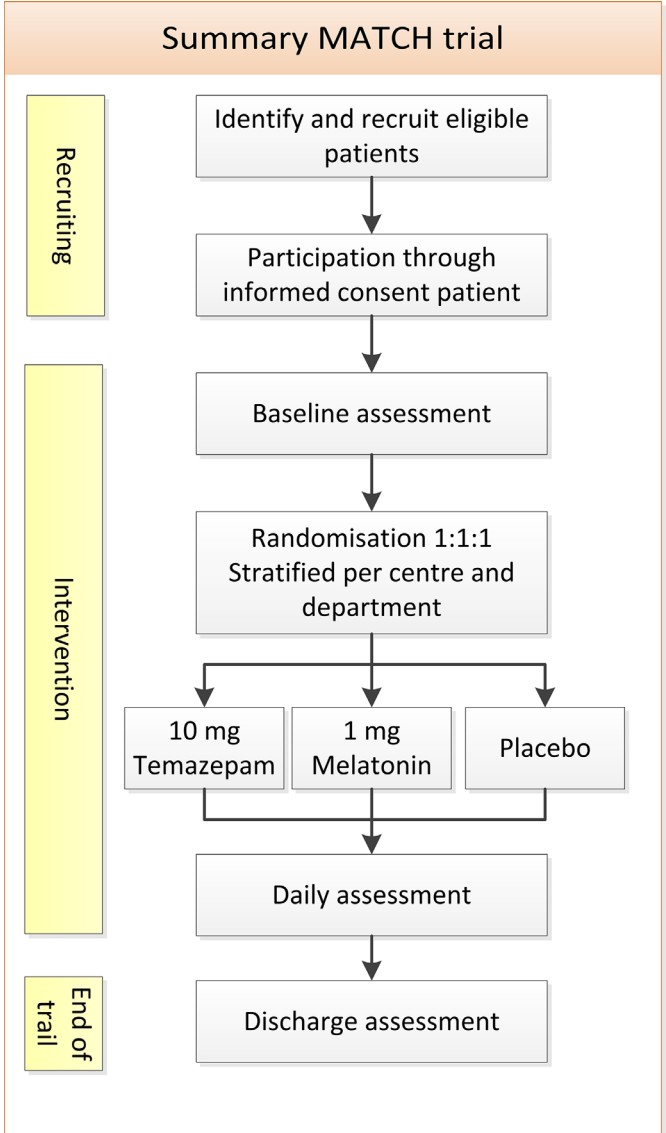

**Figure 1** Flowchart of trial design summary.

## METHODS

### Trial design

This study is a multicentre, randomised, placebo-controlled trial. Figure 1 shows the trial design summary. This study will be conducted in two university teaching hospitals and in two teaching hospitals in the Netherlands: the Academic Medical Centre (AMC) Amsterdam, the University Medical Centre Groningen (UMCG), Gelre Hospitals Apeldoorn and the Medical Centre Leeuwarden (MCL). The design of this trial protocol has followed the recommendations of the SPIRIT 2013 statement.[33]

### Patient and public involvement

Senior citizens were involved in several stages of this project, including the relevance of the study, developing the research question, the study design and conduct of the trial. The older representatives (Denktank 60+) partnered with us on the information material, the questionnaires and the burden of the questionnaires from a patient perspective. Once the trial has been published,

participants will be informed of the results in a study newsletter suitable for the public.

## Eligibility criteria

Eligible participants must have the following characteristics:

1. 60 years or older.
2. Admitted to the hospital for a medical or surgical reason.
3. Experiencing or complaining about new onset or aggravated sleep problems during hospital admission, for which a pharmacological intervention is needed based on the judgement of the attending physician.
4. Ability to fill out a sleep questionnaire.

People with the following characteristics are not eligible:

1. Inability to speak, understand or write Dutch.
2. Lack of decision-making capacity.
3. Previously diagnosed dementia.
4. Transferred from another hospital to one of the study centres, with insufficient information on previous use of sleep medication.
5. Expected stay in hospital of <48 hours.
6. Concurrent use of sedative for a sleeping disorder (>2 dosages in the last week), such as benzodiazepine, non-benzodiazepine (Z-drugs) or melatonin.
7. Alcohol consumption >13 units/week for women and >20 units/week for men.
8. Drug interactions with melatonin or contraindications for benzodiazepine use.

First selection of eligible patients will be done by the attending physician, if eligible the physician will inform the research team. Final selection will be done by a member of the research team, who also will recruit eligible participants and include them after written informed consent for participation has been obtained.

## Randomisation and blinding

After patients have given informed consent, participants will be allocated randomly to either temazepam, melatonin or placebo capsules (ratio is 1:1:1). Randomisation is stratified per centre and department, the strata include the four centres (AMC, Gelre hospitals Apeldoorn, MCL and UMCG) and departments (surgical vs medical). A stratified block randomisation made by an independent statistician is used and generated by the programme CASTOR Electronic Data Capture (Castor EDC, Ciwit BV, Amsterdam, the Netherlands, 2017). The researchers are not informed about the block sizes.

All medication used in the study will appear identical to ensure true blinding of the patient, attending physician, nurse and investigators. The study medication is manufactured by Tiofarma, Oud-Beijerland, the Netherlands and will be distributed to the study sites' pharmacies. Study medication must be kept intact at all times; the capsules cannot be broken. All study medication is packed in Single Unit Supply packages, blinded and dispensed by site trial pharmacists. The dispensing pharmacist will record the patient, patient ID and assigned medication by the randomisation process. Storage, delivery, dispensing and destruction of medication will adhere to Good Clinical Practice regulations (ICH-GCP).[34]

At the end of the study, when all data are collected, the data manager will lock the database. The information of the study arm (melatonin, temazepam and placebo) will be masked by the data manager, after which analysis can be performed in a blinded way by the researchers. Randomisation method will be broken when the study team has declared the data analysis finished.

In case of emergencies or urgent situations, such as anaphylactic reactions or loss of consciousness, and knowledge of the actual treatment is essential for further patient management, the blinding of the treatment of an individual participant can be broken (unblinded). In this case, the attending physician will contact the trial pharmacist to disclose the blinding of the study treatment for this individual patient.

## Intervention

After randomisation, all participants receive the allocated medication once daily which is in the evening before sleep (ante noctem). Study medication will be administered orally starting on the day of study enrolment.

Melatonin 1 mg and temazepam 10 mg will be used as study medication. The dosage of 1 mg melatonin is chosen based on a recent systematic review, to best mimic the normal physiological circadian rhythm of melatonin by avoiding prolonged supra-physiological blood levels in case of higher dosages.[35] The dosage of 10 mg temazepam is the recommend dose for older patients.[36]

Study medication will be continued to either a maximum of 10 days, or 12 hours before hospital discharge, or after three consecutive nights of good sleep, or until participant decides to withdraw from the study. In case sleep problems do not resolve with study medication, the dosage can be doubled. This decision will be left to the patient's attending physician. Attending physicians are instructed only to consider a double dose when the patient has slept badly for two nights in a row, in case there are no environmental or treatable physical factors that could cause the poor quality of sleep (QOS) and they should take possible adverse effects of study medication into account. Double doses of study medication will be registered in the case report form and in the medical record.

Study medication will be discontinued if a condition occurs that needs treatment with benzodiazepines (eg, anxiety), if events occur that imply a higher risk of adverse events (eg, respiratory insufficiency or starting chinolones) and in case of persisting sleeping problems. Persisting sleep problems is defined as self-reported poor QOS for three consecutive nights after study medication has been doubled.

After study medication is discontinued, either due to a reason listed above or due to end of study period, treatment will be provided according to the hospital's protocol and under the responsibility of the attending

physician. Patients remain in the study and follow the study protocol, except for using study medication.

### Timing of measurement and outcome measures

Demographic data, such as age, sex, nationality, highest educational status and living situation will be collected at baseline. Medication use, blood tests 24 hours prior to study inclusion or 24 hours after study inclusion, severity of foregoing sleep problems, depression and anxiety, level of independence, health status and vulnerability (eg, fall risk) are measured at baseline. Primary and secondary outcome measures will be recorded at baseline (T0) and every day until the 10th day (T1–T10) or until discharge. Table 1 provides an overview of the timing of measurements and the outcome measures.

### Primary outcome

The primary outcome of the study is change in self-reported sleep quality during the treatment period (T0–T10). Sleep quality will be assessed with the Leeds Sleep Evaluation Questionnaire (LSEQ).[37 38]

### Secondary outcomes

1. Change in self-reported sleep measures: getting to sleep, awakening from sleep and behaviour following wakefulness during treatment period will be assessed with the other subscales of the LSEQ. These measures will be assessed on T0 until T10.
2. Objective sleep parameters. To assess objective sleep parameters participants will wear a MotionWatch eight during the study period (T0–T10). This wristwatch-like device will calculate sleep latency, sleep efficiency, number and duration of wake bouts, time awake after sleep onset in minutes based on motion of the patient.
3. Number of good night sleep. Good nights of sleep will be assessed with a numeric rating scale for QOS. On the numeric rating scale ≤4 is considered a bad night's sleep, 5–6 is neither a good or bad night's sleep and ≥7 is a good night's sleep.[39]
4. Time needed to reach improvement in self-reported sleep quality will be based on the LSEQ as well. Improvement of sleep quality is defined as the first day that an improvement of 10 mm is reached compared with baseline.
5. Adverse events defined as confusion, delirium, reduced consciousness, falls, aspiration and other additional adverse events will be monitored daily on T0–T10.
6. Length of hospital stay in days will be recorded at discharge.

To ensure consistency and optimisation of the data collection process, all members of the research team who will include patients in the study will follow a training how to conduct measurements according to Standard Operating Procedures and how to fill out the Case Report Forms. Non-adherence to the study protocol will be obviated as much as possible. However, when it occurs this will be registered by the research team.

### Sample size calculation

Sample size calculation is based on the primary outcome measure QOS, assessed with the LSEQ. To establish a clinically relevant observed effect of melatonin, temazepam or placebo a difference of 10 mm on the QOS was chosen.[40] The SD was calculated based on the SE presented in this study. To detect a mean difference of 10 mm, two-sided, with a power of 80% and significance level of α<0,05, we need at least 210 patients per group (total of 630 patients).

The expected dropout rate during study period is 5%.[41] Therefore, we should recruit at least 663 patients, with 221 patients in each study arm.

### Data analysis

Data will be analysed according to an intention-to-treat principle. Researchers are not aware of the allocation of the intervention and therefore analyses will be conducted in a blinded way. A per-protocol analysis will be conducted excluding patients that did not receive the intervention according to protocol and were incorrectly included in the study. To determine the effects of temazepam, melatonin or placebo on sleep quality, linear mixed models will be used. Analysis will be performed on change of sleep quality over time taking baseline value into account (ie, night before start of study medication). The following covariates will be added to the model: age, gender, different concurrent medication (eg, morphine), severity of disease diagnosis (Rotterdam Symptom Checklist (RSCL), pain severity (visual analogue scale, RSCL), acute or elective admission to the hospital, foregoing sleep problems (insomnia severity index), number of patients in the room, level of independency (Katz Index of Independence in Activities of Daily Living), level of anxiety (Hospital Anxiety Depression scale) and days in the hospital before inclusion. Linear mixed models will also be used to evaluate getting to sleep, awakening from sleep, behaviour following wakefulness and the objective sleep parameters measured with actigraphy. Analyses of the time needed to reach improvement of sleep quality will be done with an interval-censored survival analysis. Proportion of good nights of sleep measured with the numeric ratings scale will be analysed, when normally distributed, with multiple linear regression analysis. For the analysis of adverse events related to study medication, descriptive statistics will be used to show which adverse events occurred. Depending on the occurrence of these adverse-events and distribution, log-linear analysis, Poisson or multiple regression analysis will be used. Time to occurrence of an adverse event will be analysed with Cox regression.

### Data management

All participants are assigned a specific trial number, and all data will be entered in an electronic trial-specific database. The data entry process will be documented and produce an audit trail. The database will be stored and maintained by Castor EDC compliant with ICH-GCP

**Table 1** Overview of the timing of measurements and outcome measures

| Measurement | Description and instrument | Baseline | Daily | Discharge |
|---|---|---|---|---|
| Sleep quality | *Leeds Sleep Evaluation Questionnaire (LSEQ),*[38 44] 10-item questionnaire, self-rated, on 0–100 mm line. Validated tool for measuring changes in sleep on four domains; getting to sleep, quality of sleep, ease and awakening from sleep and alertness and behaviour following wakefulness. The Dutch version of the LSEQ is forwards and backwards translated according to the translation guidelines[45] | x | x | |
| | Numeric rating scale for overall sleep, an overall numeric rating of quality of sleep. With '0' representing a very bad night's sleep and '10' representing a very good night's sleep | x | x | |
| Sleep pattern | *The core consensus sleep diary*[46], a 10-item sleep log to gain insight in the sleep pattern of a participant | x | x | |
| Factor influencing sleep | Based on literature,[15 47–50] an overview is made of 10 influencing factors on sleep in hospitals. Participants can indicate whether these factors influenced their sleep and if so, to what extent | x | x | |
| Objective sleep parameters | *Actigraphy, MotionWatch 8,* a wristwatch-like device, worn on the non-dominant hand. The actiwatch registers motion in epochs of 30 s. Based on motion sleep latency, sleep efficiency, number and duration of wake bouts, time awake after sleep onset in minutes can be calculated | x | x | |
| Cognition | *Mini-Mental State Examination*[51], a 10-item screening tool to evaluate global cognitive function | x | x* | |
| | *Stroop Colour-Word test*[52 53], consists of three subtasks. Used to measure attention, processing speed, cognitive flexibility and working memory | x | x* | |
| | *The Trail Making Test (TMT) A&B*[54 55], consists of two subtasks, is used to measure visual search, attention, divided attention, processing speed and cognitive flexibility | x | x | |
| Assessment of delirium | *4 AT,*[56] a four-item screening tool to detect attention deficits due to delirium | x | x† | |
| | *DSM-V,*[57] the Diagnostic and Statistical Manual of the American Psychiatric Association (DSM-5) the most widely used nomenclature by clinicians and researchers for the classification of delirium | x‡ | x‡ | |
| Severity of sleep problems | *Insomnia severity index,*[58] a seven-item questionnaire, self-rated, to quantify perceived insomnia severity | x | | |
| Depression and anxiety | *Hospital Anxiety Depression scale*[59], a 14-item questionnaire, self-rated, consisting of two seven-item subscales looking at depression and anxiety | x | | |
| Level of independence | *Katz-15 Activities of Daily Living (ADL) index scale,*[60 61] the modified Katz ADL index 15-item scale, answered with 'yes' or' no' to evaluate the level of independence | x | | |
| Health status | *Rotterdam symptom check list (RSCL)*[62], a 30-item questionnaire, self-rated, to assess physical and psychological symptom | x | x§ | |
| | *Charlson comorbidity index (CCI),*[63] a classification method for comorbid conditions. It is a valid method to estimate risk of death comorbid disease in 1 year | | | x |
| Vulnerability (eg, fall risk) | Safety management system patient screening[64] | x | | |
| Adverse events | Confusion, delirium, reduced consciousness, falls, aspiration, other additional adverse events (in complication register) and length of hospital stay in days | | x | |

*On day 3 Stroop Colour-Word test and TMT are repeated with daily assessment.
†Administered if a delirium is suspected.
‡When score on 4AT deviates, detailed assessment of mental status with DSM-V to reach diagnosis of delirium.
§On day 4 RSCL is repeated with daily assessment.

regulations[34] and the European Data Protection Directive. Confidentiality of participant information will be maintained throughout the trial. Results can only be traced to participants by the researchers involved. Data will be stored for a period of 15 years after the study has ended, according to ICH-GCP regulations.[34]

### Participant safety and monitoring

This trial is considered a low-risk study. Melatonin in a dosage of 1 mg is not known to show any adverse events and is considered safe for use by older patients. Temazepam has potential adverse-effects but this medication is part of usual treatment of hospitalised older patients with sleep problems, participation in the study will not expose patients to any additional risks. Close monitoring of all possible adverse events related to study medication will take place, and in case any adverse events occur, appropriate measures will be taken in consultation with the attending physician. The Clinical Research Unit (CRU) of the AMC will conduct the monitoring of this trial in compliance with the Medical Research Involving Human Subjects Act (WMO) and ICH-GCP regulations.[34] Annual reports of the study progress will be sent to the Medical Research Ethics Committee of the Academic Medical Centre Amsterdam. As this trial is considered as a low risk study with a low risk on serious adverse events, no Data and Safety Monitoring Board (DSMB) will be installed or interim analysis performed.

### DISCUSSION

Although sleeping problems are a common complaint in hospitalised older patients, there is still no evidence based pharmacological treatment available. Most studies on (non-)pharmacological treatment of sleeping problems in older patients were performed in community dwelling older people. However, sleep quality in hospitalised elderly is influenced by other factors than community dwelling older people and might require other therapeutic regimes.[42] For example, light and noise in the room influences the QOS in hospitalised older patients and it is difficult to measure, subsequently standardise for these confounders on the ward. In order to minimise the effect of these potential confounders, we stratified randomisation, in which patients are randomised by hospitals and departments. In this way, the known and unknown confounders are randomised and equally distributed among the groups.

In this study it was not possible to measure sleep quality and quantity using polysomnography, which is the gold standard to measure sleep quality and quantity. Instead of polysomnography, we measured objective sleep parameters indirectly by using an actigraphy, in addition to the self-reported sleep outcome measures.

This study is the first large randomised controlled trial on the effects of melatonin, temazepam or placebo in hospitalised older patients, which aimed to establish an evidence based pharmacological treatment for sleeping problems in older hospitalised patients. Irrespective of the results of this trial, the outcomes can be used for developing clinical guidelines for the treatment of sleep problems among hospitalised older patients.

### ETHICS AND DISSEMINATION

The trial will be conducted in accordance with the Declaration of Helsinki 1996 and principles of good clinical practice. Any substantial protocol amendments will be submitted to the ethics committee. A register of protocol amendments will be made available in the study protocol.

The grant was an unrestricted grant from the University of Amsterdam Foundation.

Trial methods and results will be reported according to the Consolidated Standards of Reporting Trials 2010 guidelines.[43] The study findings will be published in relevant peer-reviewed journals.

### Trial status

The trial has started on 29th of January 2018.

**Author affiliations**
[1]Department of Geriatric Medicine, University of Groningen, University Medical Centre Groningen, Groningen, The Netherlands
[2]Department of Geriatric Medicine, Medical Centre Leeuwarden, Leeuwarden, The Netherlands
[3]Department of Geriatrics, Gelre Hospitals, Apeldoorn, The Netherlands
[4]Department of Internal Medicine, Geriatrics Section, Amsterdam Public Health, Academic Medical Centre, Amsterdam, The Netherlands
[5]Department of Epidemiology, University of Groningen, University Medical Centre, Groningen, The Netherlands

**Correction notice** This article has been corrected since it first published online. The open access licence type has been amended.

**Acknowledgements** The authors thank Carolien Jansen (data manager) and the senior citizens of Denktank 60+ Noord for their time, critical appraisal and advise.

**Contributors** FS and SB contributed equally to this paper. SDR is responsible for the original conception of the study and obtaining the grant. FS, SB, SJB, NS, BCvM, NvdV, LH and SDR attributed to the design of the study. FS and SB wrote the first draft of the manuscript, all authors contributed to further drafts. All authors read and approved the final draft.

**Funding** The MATCH-trial is unrestrictedly financed by the Dalderup foundation.

**Competing interests** None declared.

**Patient consent for publication** Not required.

**Ethics approval** This study was approved by the Medical Research Ethics Committee of the Academic Medical Centre Amsterdam in June 2017. Reference number: 2015_302. The executive boards of the other participating centres have provided local feasibility approval.

**Provenance and peer review** Not commissioned; externally peer reviewed.

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
