## [Reviewer comments · BMJ Open]

ARTICLE DETAILS

TITLE (PROVISIONAL)	MelAtonin, Temazepam and plaCebo in Hospitalised older patients with sleeping problems (MATCH): a study protocol of randomised controlled trial
AUTHORS	Stenveld, Fiona; Bosman, Sjanne; van Munster, Barbara; Beishuizen, Sara; Hempenius, Liesbeth; van der Velde, Nathalie; Smidt, Nynke; De Rooij, Sophia

VERSION 1 - REVIEW

REVIEWER	Salmaan Kanji The Ottawa Hospital Research Institute, Ottawa, Ontario, Canada
REVIEW RETURNED	12-Aug-2018

GENERAL COMMENTS	Thank you for the opportunity to review this protocol for publication in BMJ open. I am pleased to see active RCTs in this field. I have no major issues with the manuscript and I feel it meets the journal's requirements for a protocol. My only suggestions would be to include a paragraph addressing limitation of the study design and steps take to minimize the impact of these limitations. Some of the limitations that I think are worth addressing include: 1. Inclusion of patients who have sleeping problems at home who may take sedatives other than benzodiazepines and melatonin.2. The issue of subjective versus objective measures of sleep quality are controversial. The gold standard method for sleep quality assessment is polysomnography which is not being employed in this study. There is further reasonable evidence to suggest that patient self report does not correlate well with polysomnography. In my opinion this is the greatest limitation of this study.3. There are no plans to standardize the non-pharmacological approach to sleep in hospital (i.e., light, noise, etc)4. Absence of a DSMB or interim analysis plan for such a large trial. I also feel that a few sentences about your plan for knowledge synthesis, dissemination and clinical impact would be of value. Minor comments: 1. in the definition of "persistent sleep problems" only the "persistent" part is defined. How are "sleep problems" defined.
--

REVIEWER	Dong-Xin Wang Peking University First Hospital Beijing, China
REVIEW RETURNED	07-Oct-2018

GENERAL COMMENTS	 1. Background (P3, L49): What is the hypothesis of the authors? Melatonin superior to temazepam? 2. Eligibility-inclusion criteria (P4, L45): How to determine that a new onset or aggravated sleep problem requires intervention? Is it possible that the criteria be described more clearly? 3. Eligibility-exclusion criteria (P4, L50): Do the authors exclude critically ill patients? For example, those who are expected to die during hospitalization? 4. Randomization and blinding (P5, L47): I suggest that the unblinding protocol in case of emergency be described in a separate paragraph. 5. Intervention (P5, L55): What does "ante noctem" mean? Before sleep? At what time will the study drug be administered? 6. Intervention (P6, L20): Do the authors have rescue drugs in case the study drugs do not work or do not work well? 7. Outcome measurement and assessment points (P8, L4): Do the authors collect history of sleep disorders? 8. Outcome measurement and assessment points (P8, L8): I suggest that the potential adverse events and management methods be described in a separate paragraph. 9. Data analysis (P8, L38-P9, L19): This part need to be reviewed by a statistician. Personally, I suggest that the methods of analysis for the primary as well as secondary endpoints should be described more clearly. Is it necessary to do sensitivity analysis in a RCT with large sample size? 10. Participant safety and monitoring (P9, L41): Patients in the placebo group will not receive active medication to improve their sleep, will this increase the risk of patients in this group? 11. I suggest that the authors add a "Discussion" section to discuss the superiority and limitation of this study.
--

VERSION 1 – AUTHOR RESPONSE

Comments by reviewer 1

Reviewer Name: Salmaan Kanji

Institution and Country: The Ottawa Hospital Research Institute, Ottawa, Ontario, Canada

Please state any competing interests or state 'None declared': none declared

1. Thank you for the opportunity to review this protocol for publication in BMJ open. I am pleased to see active RCTs in this field. I have no major issues with the manuscript and I feel it meets the journal's requirements for a protocol. My only suggestions would be to include a paragraph addressing limitation of the study design and steps take to minimize the impact of these limitations.

Answer 1: We thank the reviewer for this comment and the constructive feedback. We added the following paragraph in the discussion:

Although sleeping problems are a common complaint in hospitalised older patients, there is still no evidence based pharmacological treatment available. Most studies on (non-)pharmacological

treatment of sleeping problems in older patients concern studies in community dwelling older people. However, sleep quality in hospitalised elderly is influenced by other factors than community dwelling older people and might require other therapeutic regimes. For example, light and noise in the room influences the quality of sleep in hospitalised older patients and it is difficult to measure, subsequently standardize for these confounders on the ward. In order to minimize the effect of these potential confounders, we stratified randomisation, in which patients are randomised by hospitals and departments. In this way, the known and unknown confounders are randomised and equally distributed among the groups.

In this study it was not possible to measure sleep quality and quantity using polysomnography, which is the gold standard to measure sleep quality and quantity. Instead of polysomnography, we measured objective sleep parameters indirectly by using an actigraphy, in addition to the self-reported sleep outcome measures.

2. Inclusion of patients who have sleeping problems at home who may take sedatives other than benzodiazepines and melatonin.

Answer 2: We agree with the reviewer. We specified this in concurrent use of sedative for a sleeping problem in the eligibility criteria. If a patient is taking other sedative medication for another indication than sleep problems, this will be registered and taken into account in the data analysis. We have added the following sentence (see page 6, including track changes): Concurrent use of sedative for a sleeping disorder (>2 dosages in the last week) , such as benzodiazepine, non-benzodiazepine (Z-drugs) or melatonin.

3. The issue of subjective versus objective measures of sleep quality are controversial. The gold standard method for sleep quality assessment is polysomnography which is not being employed in this study. There is further reasonable evidence to suggest that patient self-report does not correlate well with polysomnography. In my opinion this is the greatest limitation of this study.

Answer 3: We agree with the reviewer that polysomnography would be of added value for this study. However, in this study it is not possible to use polysomnography due to ill hospitalised patients included in the study and the financial resources of this project. Therefore, we have chosen to use the actiwatch instead of polysomnography to measure the sleep quality in an objective way. We have added this limitation in our discussion, see answer 1. On the other hand, in daily clinical practice sleeping medication will be prescribed for sleeping problems based on these subjective data, which makes the implementation of the results of this study in daily practise more easy.

4. There are no plans to standardize the non-pharmacological approach to sleep in hospital (i.e., light, noise, etc).

Answer 4: We are aware that light and noise in the hospital can influence the quality of sleep. Unfortunately, it is difficult to standardize these confounders. But in order to minimize this effect of these confounders, we stratified randomisation, in which patients are randomised by hospitals and departments (strata). We added this in our discussion, see answer 1.

5. Absence of a DSMB or interim analysis plan for such a large trial.

Answer 5: This study is marked as a study with low risk on adverse events. Currently, the medication used in this trial is also used in current sleeping problems. Therefore there will be no Data and Safety Monitoring Board (DSMB) installed or interim analysis preformed. We added the following sentences on page 11 (manuscript including track changes): As this trial is considered as a low risk study with a low risk on serious adverse events, no Data and Safety Monitoring Board (DSMB) will be installed or interim analysis performed.

6. I also feel that a few sentences about your plan for knowledge synthesis, dissemination and clinical impact would be of value.

Answer 6: We thank the reviewer for this comment and adapted this in the manuscript. We added the following sentence in the manuscript: The study findings will be published in relevant peer-reviewed journals.

Furthermore the following paragraph is added in the discussion (page 13, in the manuscript with track changes):

The MATCH study is the first large randomised controlled trial on the effects of melatonin, temazepam or placebo in hospitalised older patients, which aimed to establish an evidence based pharmacological treatment for sleeping problems in older hospitalised patients. Irrespective of the results of this trial, the outcomes can be used for developing clinical guidelines for the treatment of sleep problems among hospitalized older patients.

Minor comments:

7. In the definition of “persistent sleep problems” only the “persistent” part is defined. How are “sleep problems” defined.

Answer 7: We added the following sentence: Persistent sleep problems is defined as self-reported poor quality of sleep for three consecutive nights after study medication has been doubled. (see page 8 of the revised manuscript with track changes).

Comments by reviewer 2:

Reviewer Name: Dong-Xin Wang

Institution and Country: Peking University First Hospital, Beijing, China

Please state any competing interests or state ‘None declared’: None declared.

1. Background (P3, L49): What is the hypothesis of the authors? Melatonin superior to temazepam?

Answer 1: We noticed that the hypothesis is not (clearly) reported. Therefore we added the following sentences on page 4 of the revised manuscript: In this study, we will firstly hypothesize that melatonin and temazepam are more effective than placebo on sleep quality among hospitalised older patients with sleeping problems and secondly melatonin is more effective than temazepam.

2. Eligibility-inclusion criteria (P4, L45): How to determine that a new onset or aggravated sleep problem requires intervention? Is it possible that the criteria be described more clearly?

Answer 2: We made the inclusion criteria more clearly with the following sentence: Experiencing or complaining about new onset or aggravated sleep problems during hospital admission, for which a pharmacological intervention is needed based on the judgement of the attending physician.

3. Eligibility-exclusion criteria (P4, L50): Do the authors exclude critically ill patients? For example, those who are expected to die during hospitalization?

Answer 3: We include (critical) ill patients on medical and surgical wards if they are eligible according to the eligibility criteria.

4. Randomization and blinding (P5, L47): I suggest that the unblinding protocol in case of emergency be described in a separate paragraph

Answer 4: We thank the reviewer for this useful suggestion. We fully agree and have now added in more detail when and how the blinding can be broken. The following sentences are added: "In case of emergencies or urgent situations, such as anaphylactic reactions or loss of consciousness, and knowledge of the actual treatment is essential for further patient management, the blinding of the treatment of an individual participant can be broken (unblinded). In this case, the attending physician will contact the trial pharmacist to disclose the blinding of the study treatment for this individual patient." (page 7)

5. Intervention (P5, L55): What does "ante noctem" mean? Before sleep? At what time will the study drug be administered?

Answer 5: We apologize that this expression is unclear. The term "ante noctem" means "in the evening, before sleep". We have clarified this expression in the manuscript. The following sentence is added: "After randomisation, all participants receive daily the allocated medication which is in the evening before sleep (ante noctem)." (page 7)

6. Intervention (P6, L20): Do the authors have rescue drugs in case the study drugs do not work or do not work well?

Answer 6: In case of persisting sleep problems for two nights in a row, despite of study medication, study medication can be doubled. If this is ineffective for three consecutive nights the study medication will be discontinued and a patient will be treated according to the hospital's protocol and under the responsibility of the attending physician. This was already mentioned on page 8, see the following section: "In case sleep problems do not resolve with study medication, the dosage can be doubled. This decision will be left to the patient's attending physician. Attending physicians are instructed only to consider a double dose when the patient has slept badly for two nights in a row, in case there are no if there are no other environmental or treatable physical factors that could cause the poor quality of sleep and they should take possible adverse effects of study medication into account. Double doses of study medication will be registered in the case report form and in the medical record.

Study medication will be discontinued if a condition occurs that needs treatment with benzodiazepines (e.g. anxiety), if events occur that imply a higher risk of adverse events (e.g. respiratory insufficiency or starting chinolones) and in case of persisting sleeping problems. Persisting sleep problems is defined as self-reported poor quality of sleep for three consecutive nights after study medication has been doubled.

After study medication is discontinued, either due to a reason listed above or due to end of study period, treatment will be according to the hospital's protocol and under the responsibility of the attending physician. Patients remain in the study and follow the study protocol, except for using study medication".

7. Outcome measurement and assessment points (P8, L4): Do the authors collect history of sleep disorders?

8. Outcome measurement and assessment points (P8, L8): I suggest that the potential adverse events and management methods be described in a separate paragraph.

Answer 7 & 8: After reading these comments we realized that the section on outcome measurements and assessment points were not very clear. We have adjusted the paragraph to indicate the primary outcomes, secondary outcomes and baseline measurements more clearly. See adjusted paragraph below and on page 8 and 9 of the revised manuscript with track changes.

Timing of measurement and outcome measures

Demographic data, such as age, sex, nationality, highest educational status and living situation will be collected at baseline. Medication use, blood tests 24 hours prior to study inclusion or 24 hours after study inclusion, severity of foregoing sleep problems, depression and anxiety, level of independence, health status and vulnerability (e.g. fall risk) are measured at baseline. Primary and secondary outcome measures will be recorded at baseline (T0) and every day until the 10th day (T1-T10) or until discharge. Table 1 provides an overview of the timing of measurements and the outcome measures.

Primary outcome:

The primary outcome of the study is change in self-reported sleep quality during the treatment period (T0-T10). Sleep quality will be assessed with the Leeds Sleep Evaluation Questionnaire (LSEQ).

Secondary outcomes:

1. Change in self-reported sleep measures: Getting to sleep, awakening from sleep and behaviour following wakefulness during treatment period will be assessed with the other subscales of the LSEQ. This measures will be assessed on T0 until T10.
 2. Objective sleep parameters. To assess objective sleep parameters participants will wear a MotionWatch 8 during the study period (T0-T10). This wristwatch-like device will calculate sleep latency, sleep efficiency, number and duration of wake bouts, time awake after sleep onset in minutes based on motion of the patient.
 3. Number of good night sleep. Good nights of sleep will be assessed with a numeric rating scale for quality of sleep. On the numeric rating scale ≤ 4 is considered a bad night's sleep, 5-6 is neither a good or bad night's sleep and ≥ 7 is a good night's sleep.
 4. Time needed to reach improvement in self-reported sleep quality will be based on the LSEQ as well. Improvement of sleep quality is defined as the first day that an improvement of 10mm is reached compared to baseline.
 5. Adverse events defined as confusion, delirium, reduced consciousness, falls, aspiration and other additional adverse events will be monitored daily on T0-T10.
 6. Length of hospital stay in days will be recorded at discharge.
 9. Data analysis (P8, L38-P9, L19): This part need to be reviewed by a statistician. Personally, I suggest that the methods of analysis for the primary as well as secondary endpoints should be described more clearly. Is it necessary to do sensitivity analysis in a RCT with large sample size?
- Answer 9: A statistician reviewed the data analysis part. Furthermore pointing out the need of a sensitivity analysis is correct. We choose to do the sensitivity analysis because this is the first study on sleeping problems in the hospital and we do not know how many participants will participate the full 10 days. Patients will leave the study at discharge, even if this is before the 10th day. Therefore, in order to test the robustness of the data we will analyse the data at different time points (sensitivity analyses).
10. Participant safety and monitoring (P9, L41): Patients in the placebo group will not receive active medication to improve their sleep, will this increase the risk of patients in this group?

Answer 10: There is no increased risk for patients receiving placebo. As the review van Kanji et al [1] shows, there is currently insufficient evidence that pharmacotherapy improves sleep quality and quantity in hospitalised patients. Patients receiving placebo could experience the same effect as patient receiving active treatment.

11. I suggest that the authors add a "Discussion" section to discuss the superiority and limitation of this study.

Answer 11: We agree with this point made. We added a discussion section to the manuscript. See the discussion below and in the revised manuscript with track changes on page 13.

DISCUSSION

Although sleeping problems are a common complaint in hospitalised older patients, there is still no evidence based pharmacological treatment available. Most studies on (non-)pharmacological treatment of sleeping problems in older patients concern studies in community dwelling older people. However, sleep quality in hospitalised elderly is influenced by other factors than community dwelling older people and might require other therapeutic regimes[64]. For example, light and noise in the room influences the quality of sleep in hospitalised older patients and it is difficult to measure, subsequently standardize for these confounders on the ward. In order to minimize the effect of these potential confounders, we stratified randomisation, in which patients are randomised by hospitals and departments. In this way, the known and unknown confounders are randomised and equally distributed among the groups.

In this study it was not possible to measure sleep quality and quantity using polysomnography, which is the gold standard to measure sleep quality and quantity. Instead of polysomnography, we measured objective sleep parameters indirectly by using an actigraphy, in addition to the self-reported sleep outcome measures.

The MATCH study is the first large randomised controlled trial on the effects of melatonin, temazepam or placebo in hospitalised older patients, which aimed to establish an evidence based pharmacological treatment for sleeping problems in older hospitalised patients. Irrespective of the results of this trial, the outcomes can be used for developing clinical guidelines for the treatment of sleep problems among hospitalized older patients.

VERSION 2 – REVIEW

REVIEWER	Salmaan Kanji The Ottawa Hospital Research Institute
REVIEW RETURNED	29-Dec-2018

GENERAL COMMENTS	All my concerns have been adequately addressed in this revision.
--

REVIEWER	Dong-Xin Wang Peking University First Hospital, Beijing, China
REVIEW RETURNED	27-Feb-2019

GENERAL COMMENTS	Thank you for giving me the opportunity to review the revised version. The authors have improved their manuscript. I only have some minor comments. 1. Please provide the information of trial registration. I do not find the related information in the manuscript.
--

	2. There are many acronyms in the manuscript. Please give the full name of each acronym at the first time of its appearance. 3. Page 5, line 5: “Senior citizens were involved in several stages of the is project, ...”. Do the authors mean “this project”? 4. Page 7, lines 25-29: “After study medication is discontinued, either due to a reason listed above or due to end of study period, treatment will be according to the hospital’s protocol and under the responsibility of the attending physician.” Consider the following express: “..., treatment will be provided according to ...”. 5. Page 12, lines 9-12: “Most studies on (non-)pharmacological treatment of sleeping problems in older patients concern studies in community dwelling older people.” Please consider the following express: “Most studies on (non-)pharmacological treatment of sleeping problems in older patients were performed in community dwelling older people.”
--	--

VERSION 2 – AUTHOR RESPONSE

Reviewer 2: (Dong-Xin Wang)

1. Please provide the information of trial registration. I do not find the related information in the manuscript.

Answer 1: The trial is registered in the Netherlands Trial Register (www.trialregister.nl). The trial registration number is mentioned in the last sentence of the abstract. See below:

Trial registration number: NTR6908

2. There are many acronyms in the manuscript. Please give the full name of each acronym at the first time of its appearance.

Answer 2: After this comment we check the manuscript for the acronyms and realized that we did not give the full name for all the acronyms. We adjusted this in the manuscript.

3. Page 5, line 5: “Senior citizens were involved in several stages of the is project, ...”. Do the authors mean “this project”?

Answer 3: Thank you for pointing this out. We changed the sentence in:

Senior citizens were involved in several stages of this project, including the relevance of the study, developing the research question, the study design and conduct of the trial.

4. Page 7, lines 25-29: “After study medication is discontinued, either due to a reason listed above or due to end of study period, treatment will be according to the hospital’s protocol and under the responsibility of the attending physician.” Consider the following express: “..., treatment will be provided according to ...”.

Answer 4: We changed this sentence according to the suggestion of the reviewer in:

After study medication is discontinued, either due to a reason listed above or due to end of study period, treatment will be provided according to the hospital’s protocol and under the responsibility of the attending physician.

5. Page 12, lines 9-12: “Most studies on (non-)pharmacological treatment of sleeping problems in older patients concern studies in community dwelling older people.” Please consider the following

express: "Most studies on (non-)pharmacological treatment of sleeping problems in older patients were performed in community dwelling older people."

Answer 5: We changed this sentence according to the suggestion of the reviewer in:

Most studies on (non-)pharmacological treatment of sleeping problems in older patients were performed in community dwelling older people.